# Ca^2+^-Triggered Coelenterazine-Binding Protein Renilla: Expected and Unexpected Features

**DOI:** 10.3390/ijms24032144

**Published:** 2023-01-21

**Authors:** Alexander N. Kudryavtsev, Vasilisa V. Krasitskaya, Maxim K. Efremov, Sayana V. Zangeeva, Anastasia V. Rogova, Felix N. Tomilin, Ludmila A. Frank

**Affiliations:** 1Institute of Biophysics, Federal Research Center “Krasnoyarsk Science Center SB RAS”, 660036 Krasnoyarsk, Russia; 2School of Fundamental Biology and Biotechnology, School of Non-Ferrous Metals and Material Science, Siberian Federal University, pr. Svobodny 79, 660041 Krasnoyarsk, Russia; 3Kirensky Institute of Physics, Federal Research Center “Krasnoyarsk Science Center SB”, 660036 Krasnoyarsk, Russia

**Keywords:** Ca^2+^-triggered coelenterazine-binding protein, coelenterazine, furimazine, luciferase NanoLuc, B3LYP, TDDFT, fragmented molecular orbitals method, DFTB3

## Abstract

Ca^2+^-triggered coelenterazine-binding protein (CBP) is a natural form of the luciferase substrate involved in the Renilla bioluminescence reaction. It is a stable complex of coelenterazine and apoprotein that, unlike coelenterazine, is soluble and stable in an aquatic environment and yields a significantly higher bioluminescent signal. This makes CBP a convenient substrate for luciferase-based in vitro assay. In search of a similar substrate form for the luciferase NanoLuc, a furimazine-apoCBP complex was prepared and verified against furimazine, coelenterazine, and CBP. Furimazine-apoCBP is relatively stable in solution and in a frozen or lyophilized state, but as distinct from CBP, its bioluminescence reaction with NanoLuc is independent of Ca^2+^. NanoLuc turned out to utilize all the four substrates under consideration. The pairs of CBP-NanoLuc and coelenterazine-NanoLuc generate bioluminescence with close efficiency. As for furimazine-apoCBP-NanoLuc pair, the efficiency with which it generates bioluminescence is almost twice lower than that of the furimazine-NanoLuc. The integral signal of the CBP-NanoLuc pair is only 22% lower than that of furimazine-NanoLuc. Thus, along with furimazine as the most effective NanoLuc substrate, CBP can also be recommended as a substrate for in vitro analytical application in view of its water solubility, stability, and Ca^2+^-triggering “character”.

## 1. Introduction

Ca^2+^-triggered luciferin-binding protein is one of the three key proteins (luciferase enzyme, luciferin-binding protein, and green fluorescent protein or GFP) of the coral Renilla bioluminescent system [1]. It is a noncovalent complex of the protein (single-chain polypeptide, 21 kDa) and a luciferin, coelenterazine molecule (CTZ). To be more specific, this protein was termed coelenterazine-binding protein (CBP). The polypeptide chain includes three Ca^2+^-binding sites, and here the attachment of metal ions causes the exposure of the coelenterazine-containing cavity to the external solvent, resulting in CTZ availability for oxidation in the CBP-luciferase complex with the emission of light [2]. Physiologically, the dependence of bioluminescence on Ca^2+^ connects it with nervous excitation, which ensures its protective, predatory, communicative, etc., functions.

The bioluminescent systems of most luminous marine organisms known today are based on the activity of CTZ-dependent luciferases. As a rule, these are small single-stranded proteins (20–35 kDa) that catalyze a simple reaction of CTZ substrate oxidation with blue light emission. CTZ-dependent luciferases have attracted attention, as they have been found to be excellent reporters in analytical systems for basic research, biotechnology, and biomedicine [3,4]. The genes of many of them have been cloned, and their recombinant analogues as well as artificial variants with new useful properties have been obtained. CTZ is dissolvable in organic solvents, oxidizes rapidly in air, and is recommended for use only freshly prepared, which limits its analytical application. Hence, a number of CTZ variants have been synthesized and studied with the aim of obtaining more stable derivatives as well as obtaining derivatives providing bioluminescence with alternative spectral and kinetic characteristics (see, e.g., [5,6]). On the other hand, coelenterazine ligated to CBP avoids the early auto-oxidation that creates an undesirable background luminescence. Unlike coelenterazine, CBP is stable when stored in aqueous solutions, frozen, and lyophilized [7]. Therefore, it successfully replaced CTZ in tick-borne encephalitis immunoassay [7]; the Renilla-CBP system in combination with Ca^2+^-regulated photoprotein obelin was applied for simultaneous determinations of two targets due to common dependence of bioluminescence on Ca^2+^ [8]; CBP was shown to be an effective substrate for copepod *Metridia longa* luciferase as well [9]. The increase in the bioluminescent reaction efficiency was found in both Renilla and Metridia luciferase cases [9,10]. It was found that CBP ligates not only native coelenterazine, but also coelenterazine-v, producing a stable and efficient substrate that shifts bioluminescence to the longwave region by about 40 nm [11].

NanoLuc is an artificial CTZ-dependent luciferase generated from a small subunit of the shrimp *Oplophorus gracilirostris* luciferase [12]. It became a very popular reporter due to the unusually bright and long-lasting bioluminescence, small size (19 kDa), a wide range of pH insensitivity, and enhanced stability. Furimazine (2-furanylmethylenedeoxycoelenterazine, FMZ), one of the CTZ analogues, was shown to be an optimized NanoLuc substrate, the use of which provides a significantly higher bioluminescence signal. Like CTZ, furimazine is a lipophilic molecule with a significantly limited solubility in aqueous solutions which is prone to auto-oxidation. We hypothesized that furimazine ligated to apoCBP would become water-soluble, more stable, Ca^2+^-dependent, and an effective substrate of NanoLuc luciferase in terms of bioluminescence characteristics. In this work, we have obtained and studied the furimazine-apoCBP in comparison with the other substrates aiming to find the one optimal for application in NanoLuc-based assay.

## 2. Results

### 2.1. Theoretical Evaluations

Unlike coelenterazine’s chemical structure, the structure of furimazine includes 2-furanyl radical instead of p-hydroxybenzene and 6-phenyl radical instead of p-hydroxyphenyl (Figure 1a). Furimazine molecule is slightly smaller (381 versus 423 Da), allowing it to fit into the coelenterazine cavity of the CBP (Figure 1b). However, the loss of some functional groups can lead to a decrease in stability of the ligand–protein complex. The coelenterazine-binding protein (CBP) was optimized to estimate the binding energy between the ligand and the protein. The overall spatial structure of the recombinant selenomethionine-labeled CBP, determined at 1.7 Å, approximates the protein scaffold characteristic of the class of Ca^2+^-regulated proteins (PDB code 2HPS). The crystal structure of the substrate, coelenterazine, was transformed into furimazine by manually replacing the -OH group with a hydrogen atom at R_1_ and the phenolic ring at R_3_ with a furan ring (Figure 1a). The FMZ molecule was placed into the active center of apoCBP, and the structures were optimized by FMO2-DFTB3. The absorption spectra for FMZ in the protein cavity were calculated by FMO1-TD-B3LYP/6-31G*/PCM. The calculated maximum in the absorption spectrum of FMZ in the protein cavity is 437 nm; this value is in a good agreement with the experimental one (Appendix A).

The equilibrium structures of the protein–ligand complexes are shown in Figure 2. This analysis was performed for CTZ-apoCBP and FMZ-apoCBP complexes. For CTZ, the total binding energy to the protein is −55.7 kcal/mol. When the CTZ substrate is replaced by FMZ, the ligand- and protein-binding energy decreases to −44.2 kcal/mol. The change in protein-binding energy during substrate replacement may be due to the fact that the hydrogen net between the ligand and amino acids of the protein active center is disrupted.

There are several hydrogen bonds that stabilize CTZ in the protein active center; some of them are present in the case of FMZ, but some are lost. The hydrogen bond is saved between: Arg22 (1.9 Å) and H_2_O (water molecule in the 2hps.pdb sequence No. 189) (2.7 Å) and the oxygen of the CTZ imidazole ring (1.7 Å); Tyr181 and the nitrogen (N1) of the CTZ imidazole ring through the -OH group (2.6 Å and 2.7 Å); Phe180 and the hydrogen (N7) of the CTZ imidazole group (1.8 Å and 1.8 Å). The bond is lost between the -OH-group of R_1_ and H_2_O (water molecule in the 2hps.pdb sequence No. 226) (2.0 Å) and between the -OH-group of R_3_ and Tyr36 (1.8 Å) and Arg19 (1.9 Å). The loss of several hydrogen bonds due to ligand substitution may affect the lowering of the ligand–apoprotein binding energy, i.e., furimazine should bind weaker than coelenterazine.

### 2.2. Experimental Results

#### 2.2.1. Furimazine-apoCBP Complex Preparation and Properties

The apoCBP binds furimazine very rapidly from the Ca^2+^-free solution, similar to the case with coelenterazine [10]. However, Amicon centrifugal filters were used to remove a small furimazine excess and replace the buffer solution, since the FMZ-apoCBP complex was found to be prone to dissociation upon chromatographic purification. The absorption spectrum of the homogeneous FMZ-apoCBP displays two main maxima at 275 and 447 nm and minor maxima at 360 nm (Figure 1c). The 275/447 nm absorption ratio is 2.66, close to that of the recombinant CTZ-apoCBP (2.65, [10]). The absorbance maximum of furimazine bound with protein is red-shifted from that in buffered solution (400 nm), which might be conditioned by the more hydrophobic environment of furimazine within the protein. As one may see, FMZ-apoCBP and CBP spectra obtained under the same conditions are practically identical, which implies the identical construction of the chromophore electronic system (Appendix A). In the presence of Ca^2+^, the visible maximum is shifted by 24 nm from 447 to 423 nm but not to 400 nm, corresponding to the absorbance maximum of free furimazine prepared in the same solution (Figure 1c). This shift is accompanied with a slow extinction decrease.

The protein was relatively stable when stored in a solution of 20 mM TrisHCl pH 7.0, 5 mM EDTA, frozen, and lyophilized, judging by the NanoLuc bioluminescence with FMZ-apoCBP freshly prepared or in 45 days of storage (Table 1). The decrease in activity during storage in solution occurs through the dissociation of the FMZ-apoCBP complex followed by FMZ auto-oxidation, which was detected by its absorption spectrum; the peak at 443 nm decreased, whereas the peak at 330 nm appeared, which can be attributed to oxidized furimazine (Figure 1c). When stored frozen or freeze-dried with the addition of BSA, the protein retained 85–90% of its initial substrate capacity. Note that furimazine in aqueous media completely loses its substrate ability within 2 days (100 nM in 20 mM TrisHCl pH 7.0, 8 °C).

#### 2.2.2. Bioluminescence NanoLuc Assay

Luciferase Nano Luc preparation was obtained as described in Materials and Methods and was highly purified according to the SDS-PAGE analysis (Appendix A).

Figure 3 shows bioluminescent signal records for NanoLuc with four variants of luciferins—“free” CTZ and FMZ with the bound ones—CBP and FMZ-apoCBP under the same measurement conditions (substrate and luciferase concentrations, pH, temperature, buffer). In all cases, bioluminescent signals are characterized by a rapid rise to a maximum and then a long-lasting uniform glow. The largest 5 s integral signal was observed in the case of FMZ application. The other substrates give a slightly lower signal: CBP by 21 ± 0.5%, FMZ-apoCBP by 30 ± 1.5%, and CTZ by 38 ± 2.5. At the same time, we found that NanoLuc/FMZ-apoCBP bioluminescence practically does not depend on Ca^2+^; the difference in integral signals in the presence or absence of Ca^2+^ does not exceed 10%. Thus, FMZ ligated by apoCBP is available for NanoLuc-catalyzing oxidation. In contrast, for the case of CBP as a substrate, bioluminescence is strictly dependent on Ca^2+^ and no signal is observed in its absence.

Figure 4 shows the concentration dependence of the bioluminescence intensity when NanoLuc is assayed with either FMZ, FMZ-apoCBP, CTZ, or CBP, and Lineweaver–Burk graphs are plotted on the basis of these data. The initial bioluminescence intensity with FMZ is about three times higher than that with the other substrates over the whole range of substrate concentrations. From these plots, the apparent kinetic parameters of the reactions were calculated (Appendix A). Assuming that bioluminescent reactions proceed with a certain quantum yield and the actual amount of the reaction product (oxidized CTZ or FMZ molecule) does not match the amount of light emitted, the term “apparent” has been used.

Judging from the apparent k_cat_ value, the turnover number of NanoLuc with FMZ is almost three to five times as high as the one with the other substrates. NanoLuc bioluminescent reaction efficiency calculated as the k_cat_/K_M_ ratio for FMZ is twice as high as that for the other substrates (4.5 × 10^5^ versus 1.8–2.1 × 10^5^ µM^−1^s^−1^).

It is notable that, contrary to our expectations, furimazine ligated with apoCBP does not increase NanoLuc bioluminescence compared to the “free” furimazine. We have previously observed the effect of such an increase for Renilla and Metridia luciferases in the cases of CBP and “free” coelenterazine [9,10].

The bioluminescence spectra obtained with free FMZ and FMZ-apoCBP (maxima at 450 and 453 nm, respectively) are slightly displaced when compared with the spectra obtained with coelenterazine and with CBP (maxima at 454 and 456 nm) (Appendix A). This difference can be accounted for by small differences in the structure of the emitters and also by the fact that the same type of emitter encounters a slightly different perturbation of its excited energy level in the environment.

## 3. Discussion

Of great interest is the CBP universal ability to serve as a Ca^2+^-dependent substrate for luciferases from various marine organisms: Coelenterates (soft corals Renilla) and copepods (Metridia). In both cases, the increase in bioluminescence efficiency with CBP indicates that Ca^2+^ binding does not cause the release of coelenterazine into the medium. It is rather that the formation of an intermolecular complex of luciferase-CBP-3Ca^2+^ takes place, as was clearly shown for Renilla [3]. Furimazine, a synthetic coelenterazine derivative, is one of the most optimal substrates for NanoLuc luciferase. Anticipating obtaining a water-soluble, stable, Ca^2+^-dependent variant of the substrate with an increased level of bioluminescence, we ligated furimazine to apo-CBP. The complex furimazine-apoCBP was obtained and studied as a NanoLuc substrate in comparison with furimazine, coelenterazine, and CBP. The protein is relatively stable in storage under different conditions. It “works” as a substrate with an effectiveness close to that of coelenterazine or CBP, but almost twice weaker than that of FMZ. Contrary to expectations, NanoLuc bioluminescence with the complex FMZ-apoCBP does not depend on Ca^2+^, e.g., furimazine is available for luciferase-catalyzing oxidation. It can be explained by the disappearance of two of the five hydrogen bonds, immobilizing coelenterazine in the apoCBP cavity, and as a consequence, the decrease in ligand–protein binding energy.

An unexpected result was obtained with the other variant of substrate—Renilla Ca^2+^-triggered coelenterazine-binding protein. Its application gives almost 20% lower integral NanoLuc bioluminescent signal compared to the most effective furimazine. At that, bioluminescence arises only in the presence of calcium ions. As calcium does not cause coelenterazine dissociation, NanoLuc bioluminescence may occur when a short-lived and low-specific complex with CBP appears. It can be assumed that the shrimp bioluminescent system includes a protein similar to CBP, connecting it with the animal nervous system.

The dependence on Ca^2+^ provides the use of CBP/NanoLuc-based reporters in combination with the reporters based on Ca^2+^-regulated photoproteins for the simultaneous detection of two targets in a sample. Thus, along with furimazine, CBP is a prospective substrate, especially for in vitro assay application.

## 4. Materials and Methods

Coelenterazine was purchased from Prolume Ltd. (Pinetop, AZ, USA) and furimazine from TargetMOL (Wellesley Hills, MA, USA). Recombinant coelenterazine-binding apoprotein (apoCBP) of the soft coral *Renilla muelleri* was obtained as described in [10]. All the other reagents purchased from Sigma-Aldrich (St. Louis, MO, USA), unless otherwise stated, were of analytical reagent grade or better.

### 4.1. NanoLuc Expression and Purification

*E. coli* BL21-CodonPlus (DE3) (Novagen, Sigma-Aldrich, St. Louis, MO, USA) transformed with plasmid pET22b-NLuc carrying the NanoLuc gene, codon-optimized for bacterial expression (Evrogen, Moscow, Russia) were cultivated at 37 °C in LB medium, containing 200 µg mL^−1^ ampicillin and induced by 1 mM IPTG after reaching an OD_590_ of 0.6–0.7, then further cultivated for 3 h. The cells harvested by centrifugation were resuspended in 1 mM TrisHCl, pH 8.0, destroyed by ultrasonication on ice, and then centrifuged. Streptomycin sulfate (1%) was added to the supernatant, and the formed precipitate was separated by centrifugation. Supernatant was dialyzed against 20 mM TrisHCl, pH 8.0, loaded onto the column Q Sepharose Fast Flow (GE Healthcare, Chicago, IL, USA), and proteins were eluted with NaCl gradient (0–0.3 M).

### 4.2. Furimazine-apoCBP Complex Production

ApoCBP sample in 6 M urea was diluted 20-fold with 5 mM EDTA, 20 mM TrisHCI pH 7.0 buffer, mixed with 1.1 molar excess of furimazine, and incubated for 1 h at 4 °C. Amicon centrifugal filters ((Merk Millipore, Burlington, MA, USA) were used to separate furimazine excess and replace the buffer solution with 25 mM NaCl, 1 mM EDTA, 20 mM TrisHCl pH 7.0).

The purity of isolated proteins was evaluated by SDS-PAGE analysis (Appendix A).

Protein concentration was measured with DC Bio-Rad(Hercules, CA, USA) protein assay kit.

### 4.3. Bioluminescence Assay Was Performed in Two Ways

(A) The 100 µL aliquots of coelenterazine or furimazine (in phosphate-buffered saline, PBS, with 5 mM EDTA, 0.05% BSA) and FMZ-apoCBP or CBP (in 20 mM TrisHCl pH 7.0, 0.15 M NaCl, 0.1 M CaCl_2_, 0.05% BSA) were placed into the cuvette in concentrations ranging from 0.01 to 10 µM. Bioluminescent signal was measured immediately after rapid injection of 10 µL NanoLuc in the same solution (final concentration—1 nM), using cuvette photometer BLM-003 (SDTB “Nauka”, Krasnoyarsk, Russia). Measurements were conducted at 20 °C maintained with the temperature-stabilized cuvette block of the photometer.

(B) The 100 µL aliquots of CBP or FMZ-apoCBP, CTZ, FMZ (100 nM in 20 mM Tris-HCl pH 7.0, 0.15 M NaCl, 5 mM EDTA, 0.05% BSA) were placed into the wells of opaque microtiter plates (Costar, Korning, NY, USA). Bioluminescent signal was measured immediately after rapid injection of 100 µL NanoLuc (final concentration was 1 nM in 100 mM TrisHCl pH 8.8, 0.1 M CaCl_2_, 0.05% BSA or in the same buffer without Ca^2+^ for the case of FMZ-apoCBP). Bioluminescence was integrated for 5 using a plate reader Mithras LB 940 (Berthold technologies, Bad Wildbad, Germany).

### 4.4. Spectral Measurements

NanoLuc bioluminescence spectra were measured with Varian Cary Eclipse spectrofluorometer (Agilent Technologies, Santa Clara, CA, USA). All measurements were carried out at room temperature in 20 mM Tris-HCl pH 7.0, 0.15 M NaCl, 1 mM EDTA. Bioluminescence was initiated either by CTZ or FMZ or by 0.1 M CaCl_2_ solution, when CBP or FMZ-apoCBP were applied as substrates. Emission spectra were corrected via the device software. All measurements were carried out at room temperature.

The absorption spectra were obtained with a UVIKON 943 Double Beam UV/VIS spectrophotometer (Kontron Instruments, Milan, Italy). The measurements were carried out at room temperature.

### 4.5. Theoretical Methods

The calculations of atomic and electronic structure of furimazine (FMZ) were performed using B3LYP functional [13] with the Solvation Model Based on Density (SMD) [14] and cc-pVDZ basis set. As shown in [15], the calculation of coelenterazine (CTZ) in methanol in SMD model was the most suitable for calculating optical properties. The absorption spectra of the substrate in the gas phase and solvents were calculated by the TD-DFT theory [16]. All calculations were performed in GAMESS [17] program.

The complex of coelenterazine-binding apoprotein (apoCBP) with coelenterazine or with furimazine was optimized by fragmented molecular orbitals (FMO) method [18] (FMO2-DFTB3) using 3ob-3-1 parameters [19,20], with the Conductor-like Polarizable Continuum Model (C-PCM) and Grimme D3(BJ) dispersion correction [21] implemented in GAMESS. Upon calculation of the complex (apoprotein and FMZ), the structure was divided into 208 fragments of coelenterazine-binding protein (PDB code 2HPS). The XRD structure (2hps.pdb) of the coelenterazine-binding protein (CBP) contains 186 amino acids and 176 water molecules. All amino acids, substrate (FMZ or CTZ), and water molecules in the active center and near the cavity where the substrate enters were taken into account in the FMO calculation (total of 208 fragments). Protein fragmentation into amino acids was carried out by a single peptide bond (C-N) according to [22,23].

The FMO/SA analysis was performed for the CBP and furimazine-apoCBP proteins obtained. Subsystem analysis (SA) for FMO is formulated to decompose the total energy into subsystem values [24]. The binding energy ΔE is the energy required to form a complex *AB* from two isolated (labeled 0) molecules *A* and *B*, with *A*, *B*, and *AB* at their respective energy minima. Calculations were performed by the RI-MP2/6-31*G level of the theory. The energy was calculated for each isolated system: (1) energy of the whole complex (*AB*—protein + ligand)—E_part-complex_ (E*_AB_*); (2) energy of isolated ligand (*A*)—E_part-isolated_ (E*_A0_*); (3) energy of isolated protein (*B*)—E_part-isolated_ (E*_B_*_0_). Then, ΔE_part_ was calculated as the difference between E_part-complex_ and E_part-isolated_ and multiplied by 627.51 (i.e., conversion from Hartree to kcal/mol). E_bind_ was calculated by adding PIE (pair interaction energy) and ΔE_part_. The energies obtained include the relaxation energy.

## Figures and Tables

**Figure 1 ijms-24-02144-f001:**
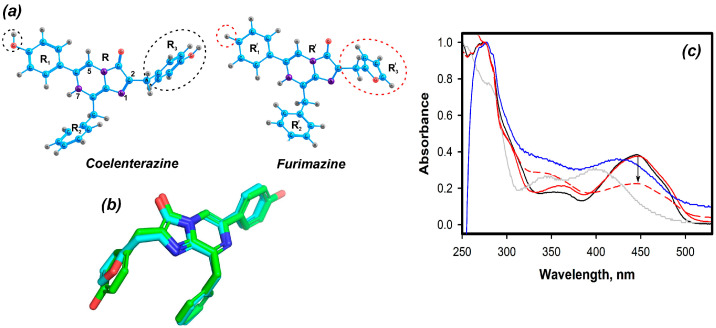
(**a**) Atomic structures of coelenterazine and furimazine substrates. Functional groups R_1_ (R’_1_), R_2_ (R’_2_), R_3_ (R’_3_), and chromophore R (R’) for CTZ and FMZ, respectively. The dotted circle shows different parts of the molecules. The C, O, H, and N atoms are colored blue, red, gray, and purple, respectively. (**b**) CTZ and FMZ superposition. (**c**) Absorbance spectra of: CBP (black line), FMZ-apoCBP (red line), FMZ-apoCBP plus Ca^2+^ (blue line), FMZ (gray line), FMZ-apoCBP after 45 days of storage in solution at 8 °C (red dotted line). Spectra are normalized at the short wavelength maximum.

**Figure 2 ijms-24-02144-f002:**
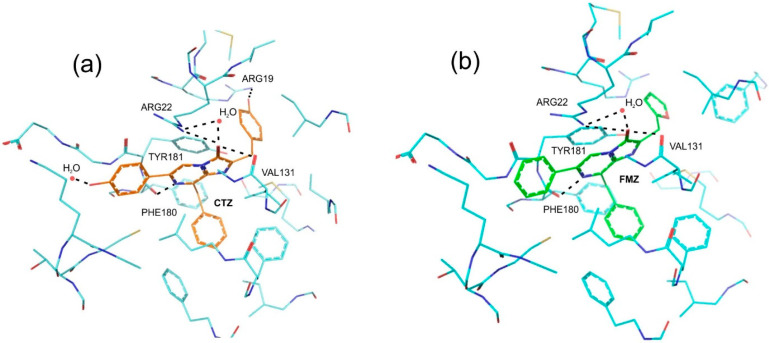
Protein active sites (PDB code 2HPS) with coelenterazine (**a**) and with furimazine (**b**). Water molecules are shown as red balls. Inferred hydrogen bonds are shown as dotted lines. The specified amino acids are involved in the formation of hydrogen bonds.

**Figure 3 ijms-24-02144-f003:**
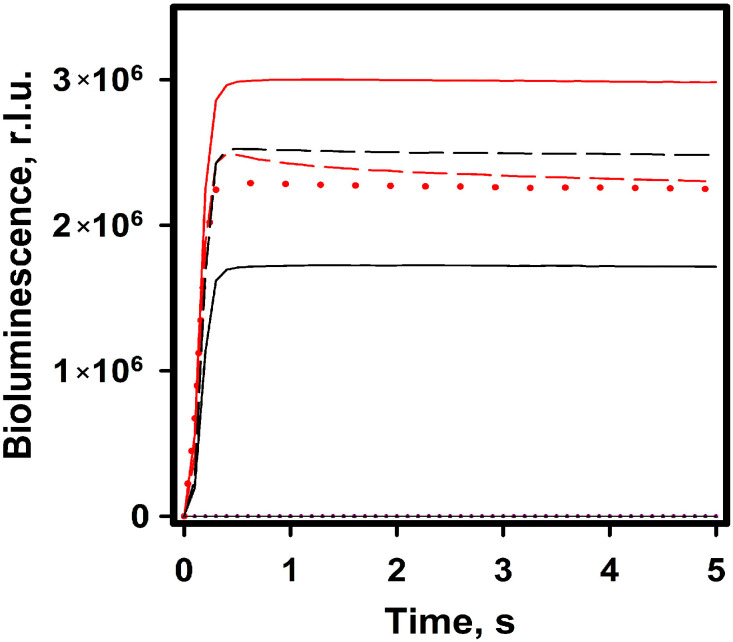
Bioluminescence signal recorded for NanoLuc luciferase with furimazine (red line), FMZ-apoCBP (red dashed line) and FMZ-apoCBP in the absence of Ca^2+^ (red point line), CTZ (black line), and CBP (black dashed line) at equal concentrations (100 nM). NanoLuc luciferase concentration was 1 nM in all experiments. The reactions were carried out in 50 mM TrisHCl pH 7.0, 0.1% BSA. Background signals from all substrates are negligible (black point lines). r.l.u.—5 s integral bioluminescence in relative light units.

**Figure 4 ijms-24-02144-f004:**
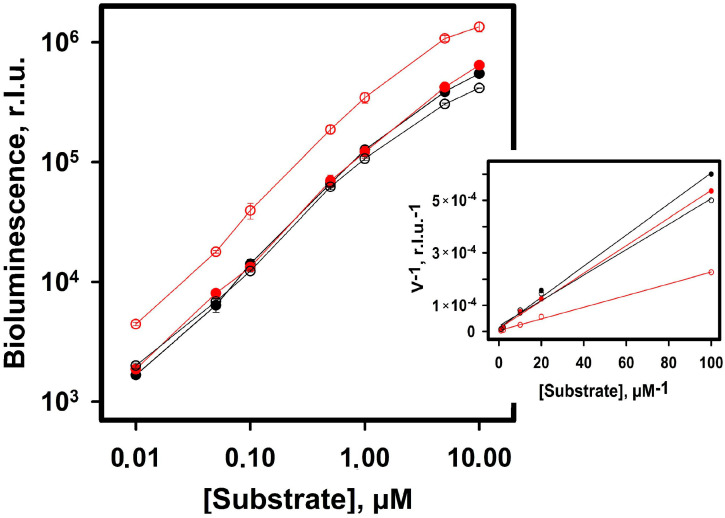
Log–log plots of NanoLuc luciferase bioluminescence assay and Lineweaver–Burk plot (inset) with furimazine (red empty circles), FMZ-apoCBP (red filled circles), coelenterazine (black empty circles), and CBP (black filled circles). The luciferase concentration is 1 nM in all experiments. The points on the plots are average of three measurements. r.l.u.—initial bioluminescence intensity in relative light units.

**Table 1 ijms-24-02144-t001:** Residual active FMZ-apoCBP as determined by NanoLuc bioluminescence.

Conditions	Freshly Prepared, %	In 45 Days, %
20 mM TrisHCl pH 7.0, 5 mM EDTA, 8 °C	100	68.4 ± 0.6
Frozen and thawed	85.8 ± 0.2	85.6 ± 0.5
Freeze-dried—dissolved	68.2 ± 1.5	62.5 ± 0.5
Freeze-dried (plus 0.1% BSA)	96.4 ± 0.3	92.7 ± 0.2

## Data Availability

Not applicable.

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
