# Peer review of "Ca2+-Triggered Coelenterazine-Binding Protein Renilla: Expected and Unexpected Features"

_ijms, 2023, doi:10.3390/ijms24032144_

Round 1

Reviewer 1 Report

The authors attempted to obtain a better substrate for NanoLuc luciferase. The aim of the manuscript was to check if furimazine (coelenterazine analog) ligated to apoCBP would become water-soluble, more stable, Ca2+-dependent and effective substrate of NanoLuc luciferase in terms of bioluminescence characteristics. Unfortunately, the new substrate turned out to be not better than the free furimazine and was not Ca2+-dependent.

The manuscript is interesting and certainly deserves the publication in IJMS journal but with minor corrections.

11.       Figure 3 does not present the dependence of CBP on Ca2+ ions. It would be desirable that it contained additional curve for CBP in the absence of Ca2+ ions.

22.       There is no information about water solubility of furimazine-apoCBP complex in comparison with free FMZ. Is it more water-soluble? Also, there is no comparison of FMZ-apoCBP complex stability with free FMZ. Did it become more stable than FMZ?

33.       The Discussion section would be more representative if you added a scheme showing interactions of FMZ-apoCBP and CBP with luciferase and dependences on Ca2+ ions.

Author Response

We are grateful to the Reviewer for their positive feedback and useful questions and comments.

Corrections made are shown in red letters.

Hope, that the revisions we made can help make the paper more clear and coherent for readers.

Below please find our answers to the questions and comments of the Reviewer 1

  1.  Figure 3 does not present the dependence of CBP on Ca2+ ions. It would be desirable that it contained additional curve for CBP in the absence of Ca2+ ions.

In case of CBP application as a luciferase (Renilla, Metridia or NanoLuc) substrate, bioluminescence is strictly dependent on Ca2+ and no signal is observed in its absence. Relevant sentence is added to the manuscript

  1. There is no information about water solubility of furimazine-apoCBP complex in comparison with free FMZ. Is it more water-soluble? Also, there is no comparison of FMZ-apoCBP complex stability with free FMZ. Did it become more stable than FMZ?

Furimazine-apoCBP complex is a protein molecule, with inherent high water solubility. FMZ is highly soluble in organic solvents (dimethyl sulfoxide). Its water solubility is extremely poor and final concentration of free FMZ is low. When stored in aqueous solution (100 nM, 20 mM TrisHCl pH 7.0, 5mM EDTA) its substrate capacity vanished in 2 days. The information was added into the text.

  1. The Discussion section would be more representative if you added a scheme showing interactions of FMZ-apoCBP and CBP with luciferase and dependences on Ca2+ ions.

In our opinion, adding such a scheme to the Discussion section will not introduce new information and will actually duplicate the content of Figure 3. However, such a scheme is very appropriate and was used by us in the graphical abstract.

Reviewer 2 Report

The authors present a study on ligand effect on bioluminiscence of luciferases. It is a work combining experimental and computational approaches. I will focus on the latter part. The work uses modern, appropriate methods to support the findings.

The results appear to be interesting for those, who may wish to develop some standardised protocols/assays involving the studied ligands. So from this point of view having deeper understanding of the properties of the complex is advantageous for future readers.

I suggest accepting the paper, after some changes reflecting the following points are introduced.

Some questions to answer

1. How did the authors fragment the protein into 208 fragments (only the apoprotein?), when the structure 2hps.pdb has 186 protein residues? I assume waters are included in the number. But at line 118 fragment No. 226 is referenced, which is more than 208.

2. The authors define how they calculated the binding energies with subsystem analysis with FMO. They define and define the “0” energies as those of isolated molecules. Yet, it is not clear whether these energies include also the relaxation energy term. Later, authors state that these were not calculated at all. Lastly, the sentence “B is the internal energy of A (protein without ligand) and B (ligand) in the AB (protein + ligand) complex” is not clear at all. So were the monomer energies calculated or not? If not, how was the binding energy evaluated? I suggest reformulating this part to gain clarity.

3. Some protein residues (amino acids) such as Val131 seem to interact with the ligand via the C=O group in the backbone. It is known, that in FMO the C=O group is miss-assigned to the neighbouring fragment. This issue is solved with the partition analysis (PA) implemented in GAMESS (see e.g. https://doi.org/10.1021/acs.jpca.0c08204, https://doi.org/10.3390/ijms232113514) where conventional residues are identical to segments (analogue to fragments). I presume the authors were cautious and accounted for the shifted C=O assignment, right?

Minor (text) modifications

I suggest reading the text and checking for small “mistakes” and typos, such as:

line 42: . ...known for today…

line 287: Grimm → Grimme

Author Response

We are grateful to the Reviewers for their positive feedback and useful questions and comments.

Corrections made are shown in red letters.

Below please find our answers to the questions and comments of the Reviewer 2

Some questions to answer

  1.  How did the authors fragment the protein into 208 fragments (only the apoprotein?), when the structure 2hps.pdb has 186 protein residues? I assume waters are included in the number. But at line 118 fragment No. 226 is referenced, which is more than 208.

The XRD structure (2hps.pdb) of the coelenterazine-binding protein (CBP) (2hps) contains 186 amino acids and 176 water molecules. All amino acids, the substrate (FMZ or CTZ), and water molecules in the active center and near the cavity where the substrate enters were taken for the calculations. The result was 208 fragments. The water fragment number in calculation #1 corresponds to the water molecule number 226 from 2hps.pdb. We left this number (226) in the paper so that the reader could understand which water molecule we are talking about.

A corresponding explanation has been added to the text in section “4.5 Theoretical methods”:

“The XRD structure (2hps.pdb) of the coelenterazine-binding protein (CBP) contains 186 amino acids and 176 water molecules. All amino acids, substrate (FMZ or CTZ), and water molecules in the active center and near the cavity where the substrate enters were taken into account in the FMO calculation (total of 208 fragments).”

  1. The authors define how they calculated the binding energies with subsystem analysis with FMO. They define and define the “0” energies as those of isolated molecules. Yet, it is not clear whether these energies include also the relaxation energy term. Later, authors state that these were not calculated at all. Lastly, the sentence “B is the internal energy of A (protein without ligand) and B (ligand) in the AB (protein + ligand) complex” is not clear at all. So were the monomer energies calculated or not? If not, how was the binding energy evaluated? I suggest reformulating this part to gain clarity.

Thanks for the comments, this paragraph has been reformulated in the article:

“The FMO/SA analysis was performed for the CBP and furimazine-apoCBP proteins obtained. Subsystem analysis (SA) for FMO is formulated to decompose the total energy into subsystem values [22]. The binding energy ΔE is the energy required to form a complex AB from two isolated (labeled 0) molecules A and B, with A, B, and AB at their respective energy minima. The energies of isolated systems (∆E=EAB–EA0–EB0). Calculations were performed by the RI-MP2/6-31*G level of the theory. The energy was calculated for each isolated system: 1) energy of the whole complex (AB - protein + ligand) - Epart-complex (EAB); 2) energy of isolated ligand (A) - Epart-isolated (EA0); 3) energy of isolated protein (B) - Epart-isolated (EB0). Then ΔEpart was calculated as the difference between Epart-complex and Epart-isolated and multiplied by 627.51 (i.e., conversion from Hartree to kcal/mol). Ebind was calculated by adding PIE (pair interaction energy) and ΔEpart. The energies obtained include the relaxation energy.”

  1. Some protein residues (amino acids) such as Val131 seem to interact with the ligand via the C=O group in the backbone. It is known, that in FMO the C=O group is miss-assigned to the neighbouring fragment. This issue is solved with the partition analysis (PA) implemented in GAMESS (see e.g. https://doi.org/10.1021/acs.jpca.0c08204, https://doi.org/10.3390/ijms232113514) where conventional residues are identical to segments (analogue to fragments). I presume the authors were cautious and accounted for the shifted C=O assignment, right?

Indeed, some amino acids (Val131, Phe180) interact with the ligand through the C=O group. In this work, protein fragmentation into amino acids was performed on a single peptide bond with regard to partitioning analysis (PA), because in FMO, during partition, carboxyls are assigned to neighboring residues, and sulfur bridges are not fragmented. The protein (PDB code 2HPS) had no sulfur bridges, which simplified the task of fragmentation. In partitioning analysis, the normal amino acid residues are identical to the segments (analogous to fragments) [https://doi.org/10.1021/acs.jpca.0c08204, https://doi.org/10.3390/ijms232113514].

A corresponding reference (No. 22 and 23) and text have been added to the section “4.5 Theoretical methods”:

“Protein fragmentation into amino acids was carried out by a single peptide bond (C-N) according to [22, 23].”

Minor (text) modifications

I suggest reading the text and checking for small “mistakes” and typos, such as:

line 42: . ...known for today… Corrected

line 287: Grimm → Grimme Corrected

The text was proofread and corrected as well.